

# Origin-destination prediction from road average speed data using GraphResLSTM model

Guangtong Hu and Jun Zhang

Capital University of Economics and Business, Beijing, China

## ABSTRACT

With the increasing demand for traffic management and resource allocation in Intelligent Transportation Systems (ITS), accurate origin-destination (OD) prediction has become crucial. This article presents a novel integrated framework, effectively merging the distinctive capabilities of graph convolutional network (GCN), residual neural network (ResNet), and long short-term memory network (LSTM), hereby designated as GraphResLSTM. GraphResLSTM leverages road average speed data for OD prediction. Contrary to traditional reliance on traffic flow data, road average speed data provides richer informational dimensions, reflecting not only vehicle volume but also indirectly indicating congestion levels. We use a real-world road network to generate road average speed data and OD data through simulations in Simulation of Urban Mobility (SUMO), thereby avoiding the influence of external factors such as weather. To enhance training efficiency, we employ a method combining the entropy weight method with the Technique for Order Preference by Similarity to Ideal Solution (TOPSIS) for key road segment selection. Using this generated dataset, carefully designed comparative experiments are conducted to compare various different models and data types. The results clearly demonstrate that both the GraphResLSTM model and the road average speed data markedly outperform alternative models and data types in OD prediction.

## INTRODUCTION

Origin-destination (OD) prediction is a pivotal aspect of urban mobility planning and intelligent transportation system (ITS), as it aids in alleviating congestion and optimizing travel efficiency during peak hours and in heavily trafficked regions (*Munizaga & Palma, 2012*). Emerging transportation solutions such as customized bus, carpooling, shared mobility services, taxi dispatching, and shared parking (*Ma et al., 2023*; *Ou & Tang, 2018*; *Narayanan & Antoniou, 2023*; *Ding et al., 2022*; *Xie et al., 2023*) are collectively referred to as on-demand mobility (ODM). ODM utilizes OD matrix to proactively optimize travel modes for passengers, thereby significantly reducing waiting times and improving overall travel efficiency. With the provision of an intelligent and flexible transportation solution, ODM not only improves user experience and service efficiency but also actively supports energy conservation (*Singh et al., 2023*). This research focuses on discovering a novel

Corresponding author
Guangtong Hu,
charles3000@cueb.edu.cn

algorithm and data type for OD prediction to cater to the growing need for efficient travel in Smart City.

Nowadays, OD prediction research primarily relies on traffic flow data collected from road segments. However, the collection of traffic flow data is often limited by equipment and resource, which means that data is only available for specific times and locations (*Tsanakas, Gundlegård & Rydergren, 2023*; *Owais, 2024*; *Bell, 1983*). It is challenging to satisfy real-time OD data requirements for non-specific locations in traffic management. Road average speed data does not require information from all vehicles and can be reliably estimated by considering only a small number of vehicles on the road, especially in congested situations where most vehicles usually travel at similar speeds. Moreover, compared to the scarcity of publicly available traffic flow data, most map APIs typically provide road average speed data. Therefore, leveraging road average speed data in OD prediction can lower the threshold for data requirements in traffic research and mitigate data inequality.

To estimate and predict OD matrix, researchers have developed numerous methodologies and models, which include traditional approaches, probabilistic models, deep learning methodologies, and various structural techniques. Several researchers have employed various models such as the least squares method (*Krishnakumari et al., 2020*), structural state-space models (*Lin & Chang, 2007*; *Zhou & Mahmassani, 2007*; *Ashok & Ben-Akiva, 2000*; *Yang, Iida & Sasaki, 1991*), maximum entropy models (*Ashok & Ben-Akiva, 2002*), and dynamic mode decomposition (*Hazelton, 2001*) to estimate and predict OD. Another category of methods relies on probability model and statistical analysis, the concept of maximum probable relative error (MPRE) (*Yang et al., 2023*), multi-criteria meta-heuristics (*Owais, Moussa & Hussain, 2019*) and encompassing Bayesian analysis (*Cheng, Trépanier & Sun, 2022*; *Tang et al., 2018*; *Vahidi & Shafahi, 2023*). Deep learning techniques have emerged as a promising field in OD prediction. Researchers have employed various deep learning architectures such as convolutional long short-term memory networks (ConvLSTM) (*Ye et al., 2023*), multi-scale convolutional LSTM (MultiConvLSTM) (*Jiang, Ma & Koutsopoulos, 2022*), spatio-temporal graph attention networks (*Chu, Lam & Li, 2020*), generative adversarial networks (GANs) (*Zhang et al., 2021*), and custom-designed deep learning frameworks (*Zhang & Xiao, 2023*; *Noursalehi, Koutsopoulos & Zhao, 2022*; *Alshehri et al., 2023*), to enhance the accuracy of OD prediction. These deep learning applications have demonstrated exceptional proficiency in addressing specific challenges inherent to OD prediction, such as sparse destination distributions (*Pamuła & Żochowska, 2023*; *Zou et al., 2022*), incomplete OD matrix (*Liu et al., 2023*), and complex Spatio-temporal environments (*Shuai et al., 2022*; *Zhang et al., 2021*; *Wang et al., 2022*; *Jin et al., 2022*), which has resulted in substantial improvements in prediction accuracy.

In this study, we employ a novel integrated framework, effectively merging the distinctive capabilities of graph convolutional network (GCN), residual neural network (ResNet), and long short-term memory network (LSTM), hereby designated as GraphResLSTM. GraphResLSTM specifically targets the in-depth exploration of spatio-temporal relationships between road network structures and average speed data along

various road segments, thereby forging a new path for accurately predicting OD matrix. The distinctiveness of the GraphResLSTM model lies in the comprehensive integration of road average speed data with a nuanced consideration of the underlying road network structure, an aspect that is often overlooked in conventional travel demand and OD prediction models. This hybrid deep learning model, merging road network structure with road average speeds, offers a fresh perspective that holds promise for significant advancements in predicting OD matrix. We conduct experiments utilizing approximately 4 million simulated vehicle records from the Huilongguan and Tiantongyuan area (commonly known as the 'HuiTian region') in Beijing, demonstrating that the proposed model outperforms other models.

The principal contributions of this article can be summarized as follows:

- Model innovation: we propose a novel hybrid model named GraphResLSTM, which presents a distinctive architecture that represents a theoretical and methodological breakthrough in the task of OD prediction;
- Data processing and source innovation: departing from the conventional reliance on traffic flow data, the study adopts road average speed data for OD prediction. This strategy not only simplifies the data acquisition process but also enhances both the accuracy and real-time nature of the predictions;
- Critical road section selection and data preprocessing methodology innovation: we employ a synthesis of the entropy weight method and the Technique for Order Preference by Similarity to Ideal Solution (TOPSIS) to identify most distinctive key road segments within the experimental road network;

The remainder of the article is organized as follows: "Data" defines the OD prediction problem and introduces the data structures and data preprocessing methods. "Road Selection" elucidates the benefits and necessity of road segment selection. In "Method", we present our proposed GraphResLSTM model. "Results and Discussion" provides experimental results based on simulated data. "Conclusions" concludes the content of this article with a summary and discussion of the findings.

## DATA

We primarily focus on the transportation system within the HuiTian region in Beijing, an area that holds a prominent position among Asia's most extensive residential communities and serves as a typical example of an urban-rural interface. The region spans approximately 168 square kilometers and accommodates a population of 1.44 million people. The total road network length is 327 km, includes one expressway, two urban expressways, 16 major urban roads, approximately 30 secondary urban roads, and over 100 urban service roads, forming a large and complex transportation system, as shown in Fig. 1 (*Changping District People's Government of Beijing Municipality, 2022*). The road network within the HuiTian region functions much like a complex multi-directional input-output system, where a multitude of roadways function as crucial access and departure points for traffic flowing in and out of the region. Within the scope of this study,

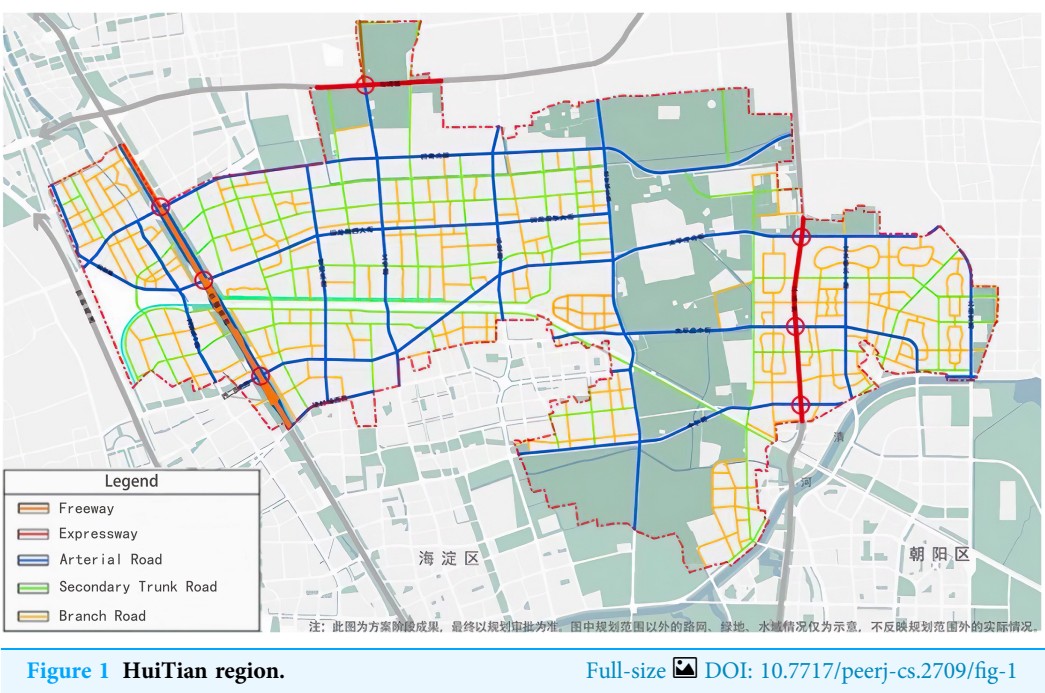

**Figure 1 HuiTian region.**

these routes are conceptualized as origin and destination points, signifying the starting and ending points of journeys.

We employ the Simulation of Urban Mobility (SUMO), a powerful platform for simulating microscopic traffic, to model the road network in the HuiTian region. SUMO offers a realistic simulation environment that replicates traffic flow conditions in the real world, thus providing a solid foundation for our study.

Following the acquisition of data from SUMO, preprocessing is essential. Due to the typically low number of vehicles present at both the start and end of the simulation phases, initial and final periods of 6,000 s each are excluded from the dataset, which are treated as the warm-up time (*Du et al., 2016*). Given the synthetic nature of the data derived from simulations, no discrepancies or errors emerge during the data cleaning process. Undertaking these preprocessing steps ensures the quality and suitability of the data, thus laying a solid foundation for the efficient training and accurate predictive capabilities of the GraphResLSTM model. We acquire simulated average speed data for each individual road segment in the area by employing this platform, which is derived by aggregating and averaging the instantaneous speeds of all vehicles on every individual road, as detailed in Table 1. This provided format offers a holistic view into the functioning status of the road network, acting as a crucial indicator for examining traffic flow and congestion trends.

In addition, we gather OD data that specifies the origin and destination of each vehicle coinciding with the time frame in which the average speed data for roads is gathered. The origins are designated at residential areas while destinations are set at entry and exit points along the roads of the region, resulting in a selection of 16 distinct OD pairs that are organized into a one-dimensional OD matrix, as exemplified in Table 2.

**Table 1 Road segment average speed data samples.**

| Data ID | Time | Road 0 | Road 1 | ... | Road 40 |
|---|---|---|---|---|---|
| 0 | 6,000 | 15.22 | 3.63 | ... | 12.61 |
| 1 | 6,060 | 14.08 | 4.87 | ... | 11.08 |
| 2 | 6,120 | 14.13 | 5.25 | ... | 13.25 |
| ... | ... | ... | ... | ... | ... |
| 1,439,800 | 86,394,000 | 0.00 | 0.00 | ... | 7.00 |

**Table 2 OD data samples.**

| OD ID | Time | OD 0 | OD 1 | ... | OD 15 |
|---|---|---|---|---|---|
| 0 | 6,000 | 417 | 216 | ... | 290 |
| 1 | 6,060 | 322 | 190 | ... | 316 |
| 2 | 6,120 | 386 | 220 | ... | 356 |
| ... | ... | ... | ... | ... | ... |
| 1,439,800 | 86,394,000 | 18 | 14 | ... | 16 |

It is worth noting that both the road average speed data and the OD data are sourced directly from the simulation without any preliminary preprocessing, thereby preserving their authenticity, ensures that our dataset maintains its integrity and offers a robust foundation for further in-depth analysis. The combined application of these datasets allows us to gain a more holistic understanding of the transportation characteristics within the HuiTian region, providing strong empirical support for future urban planning and management endeavors.

# ROAD SELECTION

In this section, we explain the advantages of road segment selection and detail the methods employed to identify key road segments.

## Necessity and advantages of road selection

With ITS constantly evolving and growing in complexity, the task of OD prediction becomes increasingly intricate due to the multitude of road segments and intersections. Consequently, the careful selection of road segments emerges as a critically important aspect (*Trivedi et al., 2024*). The advantages inherent in such a selective strategy include:

- Focus on critical areas: concentrate on areas that significantly influence overall traffic conditions, thereby gaining deeper insights and explanations of the operational characteristics of the transportation system, enhancing both depth and accuracy of the study;
- Cost-effective data collection: targeted selection of representative road segments enables us to reduce time and resource costs associated with data collection;

- Optimized resource utilization: significantly alleviate the demands on data gathering and processing, ultimately enabling a more effective allocation of finite research resources;
- Enhanced explainability and applicability: refines the model's emphasis on key road sections, which not only boosts the model's interpretability but also significantly aids in elucidating the underlying logic behind its predictions.

By concentrating attention on the most representative and influential road segments, we can maintain prediction quality while efficiently boosting computational efficiency, providing more viable solutions for real-world traffic management and planning practices.

We choose to combine the entropy weight method and the TOPSIS method for the selection of key road segments, primarily to comprehensively consider the importance of multiple evaluation indicators and ensure the objectivity and robustness of the selection process. The entropy weight method assigns weights to six key indicators—degree, clustering coefficient, degree centrality, betweenness centrality, closeness centrality, and eigenvector centrality—based on the inherent variability of the data, thus avoiding potential biases from subjective weighting. Meanwhile, the TOPSIS method identifies the most representative key road segments by calculating the distances between each segment and the positive ideal solution (PIS) and the negative ideal solution (NIS), effectively highlighting those segments that perform exceptionally well or poorly across multiple evaluation criteria.

## Entropy of road sections and the method for allocating weights

In the road segment selection process, we employ a suite of six metrics: degree, clustering coefficient, degree centrality, betweenness centrality, closeness centrality, and eigenvector centrality. We use these metrics to calculate entropy for each road segment and subsequently apply an entropy weighting method to assign weights accordingly. Here is a detailed introduction to the six key metrics:

- Degree: represents the number of other road segments connected to a given road segment. A higher degree indicates stronger connectivity of the road segment within the network. Degree is chosen as an assessment metric because it directly reflects the connectivity strength of a road segment within the network, aiding in the identification of key nodes;
- Clustering coefficient: measures the ratio of actual connections among a road segment's neighboring nodes to the maximum possible connections. The higher the clustering coefficient, the closer-knit the connections are among the neighboring road-segments. Clustering coefficient helps understand the local connectivity between road segments, which is crucial for identifying dense areas within the network;
- Degree centrality: quantifies the relative importance of a road segment in the entire network. A higher degree centrality signifies a more prominent position for the road segment in the network. Degree centrality is a fundamental centrality measure that intuitively reflects the importance of a node within the network, serving as a basic indicator for assessing node importance;

- Betweenness centrality: reflects the significance of a road segment in connecting other road segments within the network. A higher betweenness centrality implies greater control or influence over the flow of traffic within the network by the road segment. Betweenness centrality reveals the intermediary role of nodes within the network, which is essential for understanding the distribution and control of traffic flow;
- Closeness centrality: gauges how easily a road segment can reach all other road segments in the network. A higher closeness centrality means better accessibility from the road segment to the rest of the network. Closeness centrality reflects the accessibility of a node within the network, which is valuable for evaluating the position and function of a node;
- Eigenvector centrality: assesses the extent to which a road segment is linked with other important road segments in the network. Higher eigenvector centrality indicates a stronger connection between the road segment and other influential nodes in the network. Eigenvector centrality considers not only the number of connections a node has but also the importance of the nodes it is connected to, providing a more comprehensive evaluation of a node's status within the network.

In summary, these metrics are employed to evaluate the connectivity strength, local connectivity, importance, and capacity to control traffic flow of segments within the network. By integrating the aforementioned six key indicators, we can comprehensively and intricately depict the characteristics and dynamic behavior of the internal structure of the road network. Each indicator provides a unique perspective on assessing the significance of road segments; when combined, they enable us to more accurately identify which segments play a critical role in overall traffic efficiency during specific time periods.

### Entropy of road section

For each road segment, we calculate six key metrics: degree, clustering coefficient, degree centrality, betweenness centrality, closeness centrality, and eigenvector centrality. Subsequently, we employ an entropy formula to compute the entropy of the road segment, as expressed in Eq. (1). Here, the entropy value reflects the variability of a segment on a specific indicator. A lower entropy value indicates a higher information content for that indicator, implying that it is more significant in distinguishing between different road segments.

$$H(x) = - \sum_{i=1}^{n} P(x_i) \cdot log_2 P(x_i) \tag{1}$$

where $n$ represents the number of metrics, and $P(x_i)$ denotes the normalized value of the road segment in the $i\text{-}th$ metric.

### Entropy weight method

In order to consider the influence of each metric on the road segments, we utilize the entropy weighting method to assign weights, which is a well-established technique in decision-making processes involving multiple criteria. This method initiates with calculating the entropy weight for each metric and normalize these weights to arrive at the

final weights. The formula for computing the entropy weight $E(x_i)$ of a metric is given below:

$$W_i = \frac{1 - E(x_i)}{\sum\limits_{j=1}^{m}(1 - E(x_j))}. \tag{2}$$

We then calculate the weight $W_i$ for each indicator using the entropy weight formula. These weights reflect the relative importance of each evaluation metric within the overall decision-making process. Based on these weights, we performed a weighted processing of the original data, generating a weighted standardized decision matrix. Each element in this matrix represents a normalized road segment performance score that takes into account the importance of its corresponding indicator.

### Weight distribution

Upon computation of the entropy weights corresponding to each metric, we proceed to multiply these weights with the respective metric values recorded for each individual road segment. This multiplication yields a comprehensive assessment score for each road segment, which subsequently serves as its weight, accurately reflecting its comparative significance within the broader transportation network.

### Representative road section selection based on weighted TOPSIS method

In this section, we employ the TOPSIS to identify the most representative road segments based on their weights. TOPSIS assigns a comprehensive score to each road segment by assessing its proximity to both the PIS and NIS.

The PIS and NIS are established by extracting the best and worst attribute values from the decision matrix. Within our context, the PIS embodies a hypothetical road segment with the highest scores across all considered metrics, while the NIS signifies the road segment with the least favorable values in each metric. The calculation method for the *j-th* component of the PIS and NIS solutions is as follows:

$$PIS_j = \max_{i=1}^{m}(a_{ij}) \; for \; j = 1, 2, \ldots, n \tag{3}$$

$$NIS_j = \min_{i=1}^{m}(a_{ij}) \; for \; j = 1, 2, \ldots, n. \tag{4}$$

Mathematically, these vectors are represented as $PIS = [PIS_1, PIS_2, \ldots, PIS_n]$ and $NIS = [NIS_1, NIS_2, \ldots, NIS_n]$, respectively denoting the ideal and non-ideal state for all evaluation criteria.

To perform TOPSIS calculations, we initially normalize the performance metrics of each road segment. For each criterion, we standardize the road segment's value by dividing it by the maximum value of that particular criterion, thus creating a normalized decision matrix. Subsequently, we multiply each element of this normalized matrix by its corresponding entropy-based weight, generating a weighted normalized decision matrix.

After that, we use the Euclidean distance as the similarity metric to quantify the degree of similarity between each road segment and both the PIS and NIS, which produces two separate similarity matrices. Ultimately, these similarity matrices are combined, and the overall performance score for the *i-th* alternative $RC_i$ is computed according to Eq. (5):

$$RC_i = \frac{\sqrt{\sum_{j=1}^{n}(a_{ij} - NIS_j)^2}}{\sqrt{\sum_{j=1}^{n}(a_{ij} - PIS_j)^2} + \sqrt{\sum_{j=1}^{n}(a_{ij} - NIS_j)^2}} \tag{5}$$

where $a_{ij}$ signifies the element in the *i-th* row and *j-th* column of the weighted and normalized decision matrix, indicating the performance of the *i-th* alternative on the *j-th* evaluation criterion. $n$ denotes the total number of evaluation indicators.

The relative closeness coefficient $RC_i$ values that are closer to one indicate that the corresponding road segments are nearer to the ideal solution, thus making them more representative. Based on the relative closeness coefficient $RC_i$, all road segments are ranked. The top-ranked segments with the highest $RC_i$ values are selected as the most representative critical road segments. Employing this weight based TOPSIS method, there is a significant difference in influence between the top 41 road segments and the subsequent ones. We identify the top 41 road segments with the highest overall scores, directing future data procurement activities, thereby guaranteeing that the collected data is both more exhaustive and illustrative of the transportation network's characteristics.

## METHOD

In this section, we review four related popular neural network architectures, including CNN, GCN, LSTM, and ResNet, followed by a discussion of the proposed GraphResLSTM. Portions of this text were previously published as part of a preprint (*Zhang & Hu, 2024*).

### CNN

The task of predicting travel demand and OD flows necessitates spatial relationship modeling, making CNN potentially powerful tools for processing geospatial data (*Albashish, 2022*). In the context of 2D convolutional operations applied to input data for feature extraction, the mathematical expression can be expressed as Eq. (6):

$$(f * g)(t) = \sum_{a=-\infty}^{\infty} \sum_{b=-\infty}^{\infty} f(a,b) \cdot g(t-a, t-b) \tag{6}$$

$f$ represents the input data, and $g$ denotes the convolution kernel, where $f * g$ yields the output feature map.

Pooling operations serve to diminish the dimensions of feature maps, thereby decreasing computational complexity. A common mathematical expression for max pooling, which retains only the maximum value within each pooling region, is as follows:

$$MaxPooling(x, y) = \max_{i,j} x_{i,j} \qquad (7)$$

$x$ represents the input data.

It is crucial to leverage the proficiency of CNNs in identifying local patterns while recognizing their potential limitations in capturing long-term dependencies when forecasting OD flows.

## GCN

GCN demonstrates exceptional proficiency in processing complex relational data structures, particularly those characterizing the interwoven road segments within an urban transportation network (*Meng et al., 2022*). It employs nodes to represent entities such as junctions, and edges to depict their interconnections, exemplified by the connections between streets. Consequently, each node aggregates information from its neighboring nodes, thereby enhancing its encoded understanding of its positional relevance and significance within the broader network. This process is rigorously defined by Eq. (8):

$$H^{(l+1)} = f\left(\hat{D}^{-\frac{1}{2}}\hat{A}\hat{D}^{-\frac{1}{2}}H^{(l)}W^{(l)}\right). \qquad (8)$$

$H^{(l+1)}$ represents the node feature representations in the $l + 1$ layer, where $\hat{D}$ denotes the degree matrix, $\hat{A}$ is the symmetrically normalized adjacency matrix, $W^{(l)}$ stands for the weight matrix at the $l$-*th* layer, and $f$ represents the activation function.

In summary, GCN excel in processing complex relational data structures, particularly when depicting the intricate web of road segments within urban traffic networks. Nodes represent entities such as intersections, while edges signify connections between them, such as the links formed by streets. Using this structure, GCNs are capable of aggregating information from neighboring nodes, thereby enhancing their encoding of spatial relevance and understanding of the importance within the entire network.

## LSTM

LSTM, a variant of recurrent neural network (RNN), is specifically engineered to handle the challenge of long-term dependencies in sequential data (*Chen & Aleem, 2024*). Distinct from conventional RNN, LSTM employs a more complex internal architecture that effectively captures and retains long-term context information. The fundamental unit within an LSTM cell comprises three critical gating mechanisms: forget gate, input gate, and output gate.

The forget gate determines how much of the past information from the previous time step should be discarded at the current time step:

$$f_t = \sigma\left(W_f \cdot [h_{t-1}, x_t] + b_f\right). \qquad (9)$$

The input gate controls the amount of new information to be admitted into the cell:

$$i_t = \sigma(W_i \cdot [h_{t-1}, x_t] + b_i). \tag{10}$$

The candidate memory cell is computed by the tanh function:

$$\tilde{C}_t = tanh(W_C \cdot [h_{t-1}, x_t] + b_C). \tag{11}$$

Using the outputs from both the forget gate and input gate, the cell state is updated:

$$C_t = f_t \cdot C_{t-1} + i_t \cdot \tilde{C}_t. \tag{12}$$

The output gate determines the hidden state at the current time step, which is based on a weighted combination of the current input and the cell state:

$$o_t = \sigma(W_o \cdot [h_{t-1}, x_t] + b_o) \tag{13}$$
$$h_t = o_t \cdot tanh(C_t). \tag{14}$$

These gates regulate the flow of information, allowing LSTM to strategically discard irrelevant data, retain important details, and generate appropriate outputs. The gating mechanism in LSTM enables it to maintain long-term contextual information and effectively address the vanishing gradient problem, solidifying its crucial role in processing temporal data. This capability allows LSTM to learn patterns from historical traffic data, facilitating accurate predictions of OD changes.

### ResNet

ResNet is a deep learning architecture devised to mitigate the gradient vanishing and exploding issues prevalent during the training of deep neural networks (*Oyewola et al., 2021*). For a given input $x$, the output $y$ of a residual block is computed as follows:

$$y = F(x, W_i) + x. \tag{15}$$

$F(x, W_i)$ represents the residual mapping function parameterized by the set of weights $W_i$.

ResNet is designed to address the vanishing and exploding gradient problems encountered during the training of deep neural networks. For a given input $x$, ResNet introduces skip connections, also known as shortcut connections, which allow information to bypass one or more layers and be directly propagated to subsequent layers. This mechanism alleviates the gradient issues in deep networks by facilitating the flow of gradients through the network during backpropagation.

### GraphResLSTM

We are developing a hybrid model named GraphResLSTM by merging GCN, LSTM, and ResNet. GraphResLSTM exhibits several key strengths when used for predicting OD based on road average speed data.

GCN effectively captures the intricate spatial interdependencies among road segments. By using graph-based configurations, the model enhances our understanding of the interconnectedness and mutual impacts among different road segments within the traffic

network, thereby improving its responsiveness to fluctuations in traffic flow patterns. LSTM demonstrates an exceptional ability to reveal long-term dependencies in historical time series data, enabling the model to extract insights from past traffic conditions and make well-informed predictions about future variations in road average speeds. ResNet addresses the challenges of vanishing gradients and overfitting, enabling the model to efficiently analyze a wide range of features in road average speed datasets while preserving the subtleties of both spatial and temporal dependencies. Therefore, GraphResLSTM synergistically combines these three components to provide a more robust and insightful prediction framework for estimating OD matrices in urban transportation systems. The network architecture diagram is presented in Fig. 2.

# RESULTS AND DISCUSSION

In this section, we conduct a series of comparative experiments aimed at evaluating the performance of GraphResLSTM model and investigating the impact of diverse data types on OD prediction tasks. Through meticulous analysis, we evaluate the model's performance under various conditions, highlighting the distinct impacts that different data types have on OD prediction, and we perform significant tests to validate these impacts. Furthermore, we perform sensitivity analyses to gain deeper insights into the disparities in model performance and assess the suitability of various data types for OD prediction.

## Experiment design

We use the SUMO microscopic traffic simulation software to conduct traffic simulations on a real-world road network, specifically the Huitian area road network (Lu, Zhou & Zhang, 2013; Tang et al., 2021). The data collected from the traffic simulations include the traffic states of all roads in the Huitian area and the OD data for each vehicle. Effective sensor placement is crucial for ensuring the acquisition of high-quality data (Owais, 2022). All the simulation experiments in SUMO run for 1,000 days, with the first 100 min treated as the warm-up time. Each entry in the dataset encapsulates the traffic situation at a specific juncture in time, incorporating elements such as the recording timestamp, average speed, traffic volume, wait duration, travel time, stopping frequency for each road segment, and the quantity of vehicles corresponding to every OD pair. To simplify the training process, we convert the OD matrix into a one-dimensional tensor where each index represents a unique OD pair; a more detailed explanation of this transformation is provided in "Road Selection". We set the time interval at 1 min, which results in approximately 1.44 million records of traffic condition data. Our experimental setup involves allocating 80% of this data for training and reserving the remaining 20% for testing.

To evaluate the performance of our model, we adopt four commonly used metrics: mean absolute error (MAE), mean squared error (MSE), root mean squared error (RMSE), and symmetric mean absolute percentage error (SMAPE) (Lu et al., 2015). MAE offers a straightforward measure of average deviation magnitude, agnostic to error direction, with lower sensitivity to outliers, facilitating comprehension of the model's average prediction error, as shown in Eq. (16). MSE highlights the impact of larger prediction errors through

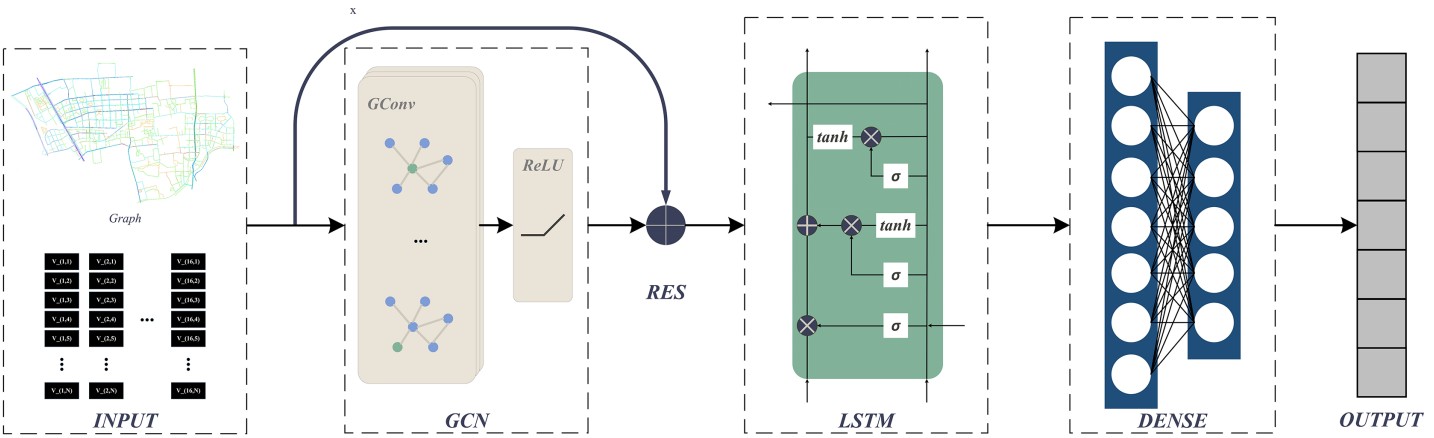

**Figure 2 GraphResLSTM.**

squared differences, illuminating the model's performance under extreme conditions, as shown in Eq. (17). As the square root of MSE, RMSE provides error magnitude in the original data units, aiding intuitive understanding of predictive precision as shown in Eq. (18). SMAPE constitutes an improved percentage error metric, circumventing issues with traditional mean absolute percentage error (MAPE) when actual values approach zero, offering a symmetric and relative error assessment suitable for datasets with broad value ranges as shown in Eq. (19). SMAPE is used to avoid division errors when the true value $y_i$ equals zero in MAPE. Collectively, utilizing these metrics ensures a holistic appraisal of model performance, encompassing absolute error size, outlier responsiveness, predictive accuracy, and relative error depiction, thereby furnishing multi-faceted evidence of the model's efficacy and reliability.

$$MAE = \frac{1}{n} \sum_{i=1}^{n} |y_i - \widehat{y_i}| \tag{16}$$

$$MSE = \frac{1}{n} \sum_{i=1}^{n} (y_i - \widehat{y_i})^2 \tag{17}$$

$$RMSE = \sqrt{\frac{1}{n} \sum_{i=1}^{n} (y_i - \widehat{y_i})^2} \tag{18}$$

$$SMAPE = \frac{100\%}{n} \sum_{i=1}^{n} \frac{|\hat{y}_i - y_i|}{(|\hat{y}_i| + |y_i|)/2}. \tag{19}$$

Here, $y_i$ and $\hat{y}_i$ denote actual and predicted values respectively, and $n$ is the sample size.

MAE provides a fundamental linear measure of error, suitable for quickly assessing the overall accuracy of the model. MSE and RMSE highlight the model's performance in handling extreme cases, aiding in the identification of potential problem areas by emphasizing larger errors. SMAPE allows for a more accurate reflection of the

proportional error relative to the actual values, providing insights into the prediction accuracy as a percentage of the true values.

All experiments in this study were conducted on a computing platform equipped with a 12th Gen Intel® Core™ i7-12700 H processor (base clock speed of 2.30 GHz) and an NVIDIA GeForce GTX 3060 GPU (driver version 560.94). The system runs on Windows 11 Home Chinese Version 23H2 and is supported by 40.0 GB of RAM. All code was written and executed using Python 3.9. This hardware configuration ensured efficient and reliable model training and evaluation, providing a solid foundation for the consistency and reproducibility of the experimental results. The standard parameter configurations for each model are presented in Table 3. Hyperparameters are being fine-tuned through an iterative process. We systematically experiment with multiple typical hyperparameter values and select those that yield the best results.

## Model evaluate

To ensure a fair comparison, identical parameter configurations are used across all models. The input sample time length is kept consistent with that of the GraphResLSTM model, and the final embedding dimension is set to 64 for all baseline models. Top-performing models on the validation set are selected for comparative analysis. For a comprehensive evaluation of the GraphResLSTM model, several classic models are chosen for comparison, such as CNN, GCN, LSTM, CNN-LSTM, and GCN-LSTM. Each model is trained and tested under the same dataset and experimental conditions, thereby ensuring an equitable and reliable assessment in the comparative study.

Table 4 shows the performance of GraphResLSTM compared to other models according to four commonly used measures: MAE, MSE, RMSE, and SMAPE. The GraphResLSTM model achieves an MAE score of 0.075958893, which is 4.34% lower than that of the LSTM model and nearly 10% below other models, signifying enhanced precision in OD prediction. With an MSE of 0.009412067, GraphResLSTM outperforms the LSTM model by 8.2% and other models by over 16%, implying greater resilience in OD prediction. When considering RMSE, GraphResLSTM model records 0.096057905. This is 3.9% lower than the GCN-LSTM model and more than 5% below other models, indicating improved consistency in OD prediction. Regarding SMAPE, GraphResLSTM model attains a score of 20.64%. Although slightly higher (by 0.68%) than the GCN-LSTM model, it remains over 6% below other models, suggesting heightened explainability in OD prediction.

Compared to other models, GraphResLSTM model demonstrates superior performance across all evaluation metrics, as shown by previous research (*Shang et al., 2020*; *Bi et al., 2021*), such adaptability is crucial for maintaining system efficiency and reliability in real-world applications. Its lowest MAE, MSE, and RMSE scores, along with an SMAPE that is just 0.68% above the minimum level, attest to its strong predictive accuracy, robustness, and consistency in OD prediction. In summary, the GraphResLSTM model currently serves as an effective, efficient, and reliable deep learning solution with exceptional performance in OD prediction using road average speed data.

**Table 3 Default values of parameters.**

| Parameters | Describe | Values |
|---|---|---|
| out_channels_gcn | Number of channels in the output data of GCN | 64 |
| hidden_size_lstm | Size of LSTM hidden layer | 64 |
| learning_rate | Learning rate | 0.00001 |
| num_epochs | Num of epochs | 1,000 |
| batch_size_train | Batch size | 512 |
| l2_loss | L2 regularization coefficient in the optimizer | 0.0005 |
| res_ratio | The residual ratio in the residual layers | 1 |

**Table 4 Models performance evaluation.**

| | MAE | MSE | RMSE | SMAPE |
|---|---|---|---|---|
| CNN | 0.082804094 | 0.011285882 | 0.10456171 | 22.37051838 |
| GCN | 0.084134158 | 0.011512537 | 0.106558977 | 22.83664327 |
| LSTM | 0.07940893 | 0.010252437 | 0.101354543 | 22.0205108 |
| CNN LSTM | 0.084023348 | 0.011509673 | 0.10608842 | 22.45860681 |
| GCN LSTM | 0.084090347 | 0.011510888 | 0.099953406 | 20.50259149 |
| GraphResLSTM | 0.075958893 | 0.009412067 | 0.096057905 | 20.64164395 |

Additionally, we conducted t-tests on the results of the model comparison experiments, as shown in Table 5. The t-test results indicate that our model has a statistically significant advantage over the other models.

## Data evaluate

The OD prediction results for five distinct input data types, presented in Table 5, illustrate how different data types affect model performance. All five data types exhibit closely similar MAEs near 0.08, indicating that the model consistently maintains a relatively low average error independent of the data type. Notably, the MAE for speed data is at 0.076, slightly lower than the others, suggesting a potentially smaller average error when predicting with speed data inputs.

The MSEs for the five data types are similarly close, approximating 0.011, denoting that the model consistently achieves a relatively low mean squared deviation in predictions for each data type. The MSE for speed data is recorded as 0.0094, a marginal decrease from the other four types, indicating a smaller mean squared deviation particularly when predicting with speed data.

The RMSEs across the five data types are uniformly low, averaging around 0.10, thereby demonstrating that the model consistently exhibits a small average absolute error in predictions across these varied data types. The RMSE for speed data sits at 0.0961, slightly under the figures for the other four data types, suggesting a potentially smaller average absolute error when using speed data for prediction.

**Table 5  t-test of models performance evaluation.**

|  | t-statistic | p-value |
|---|---|---|
| CNN | 14.42359706 | 0.00000696 |
| GCN | 15.09879891 | 0.00000532 |
| LSTM | 4.53341416 | 0.00396079 |
| CNN LSTM | 16.34171955 | 0.00000334 |
| GCN LSTM | 2.73288276 | 0.03405555 |

All five data types present SMAPE percentages around 22%, inferring that the model sustains a relatively small average relative error across these data types. Notably, the SMAPE value for speed data is measured at 20.64%, slightly lower than the others, indicating a smaller average relative error when using speed data for prediction.

In conclusion, the MAE, MSE, RMSE, and SMAPE metrics for the five data types are quite comparable, collectively demonstrating that the model displays a generally strong predictive performance. Table 6 shows the performance of road average speed data compared to other data according to four commonly used measures: MAE, MSE, RMSE, and SMAPE. Among these, the prediction precision for speed data is modestly superior to the other four data types.

Additionally, we conducted t-tests on the results of the data comparison experiments, as shown in Table 7. The t-test results indicate that our data has a statistically significant advantage over the other data.

### Sensitive analysis

We select three distinct time intervals (90–120, 1,090–1,120, and 2,090–2,120) from the testing dataset to visually depict the prediction outputs produced by GraphResLSTM, as shown in Fig. 3. In these visualizations, the upper row of graphs presents the model-generated traffic patterns, while the lower row presents the actual observed values. Across each subplot, the x-axis signifies the 16 unique OD pairs, and the y-axis represents the various time intervals within the chosen periods.

A notable observation is that there is a considerable volume of vehicle traffic between OD pair number 6 and 7, with appreciably fewer vehicles traveling *via* other OD pairs. Across different timeframes, the GraphResLSTM model generally excels in accurately forecasting the OD data for most regions. However, it is important to note that the model sometimes struggles to accurately predict OD pairs with lower volumes.

Furthermore, we conduct a sensitivity analysis on crucial hyperparameters within the GraphResLSTM model. The key parameters include: (1) the learning rate; (2) the L2 regularization coefficient in the optimizer; and (3) the residual ratio in the residual layers. The learning rate acts as a central control parameter influencing the training process, determining the magnitude of parameter adjustments at each iteration. With other parameters held constant at their defaults, we vary the learning rate across the set [1e−6, 5e−6, 1e−5, 3e−5, 5e−5]. According to Fig. 4, as the learning rate increases, four evaluation metrics tend to rise initially and then decrease, with the model exhibiting optimal
**Table 6 Data performance evaluation.**

|  | MAE | MSE | RMSE | SMAPE |
|---|---|---|---|---|
| Flow | 0.082804094 | 0.011285882 | 0.10456171 | 22.37051838 |
| Waiting | 0.084134158 | 0.011512537 | 0.106558977 | 22.83664327 |
| Travel | 0.07940893 | 0.010252437 | 0.101354543 | 22.0205108 |
| Halting | 0.084023348 | 0.011509673 | 0.10608842 | 22.45860681 |
| Speed | 0.075958893 | 0.009412067 | 0.096057905 | 20.64164395 |

**Table 7 t-test of data performance evaluation.**

|  | t-statistic | p-value |
|---|---|---|
| Flow | 33.96762516 | 0.00000004 |
| Waiting | 44.86699889 | 0.00000001 |
| Travel | 9.31258479 | 0.00008681 |
| Halting | 25.83634903 | 0.00000022 |

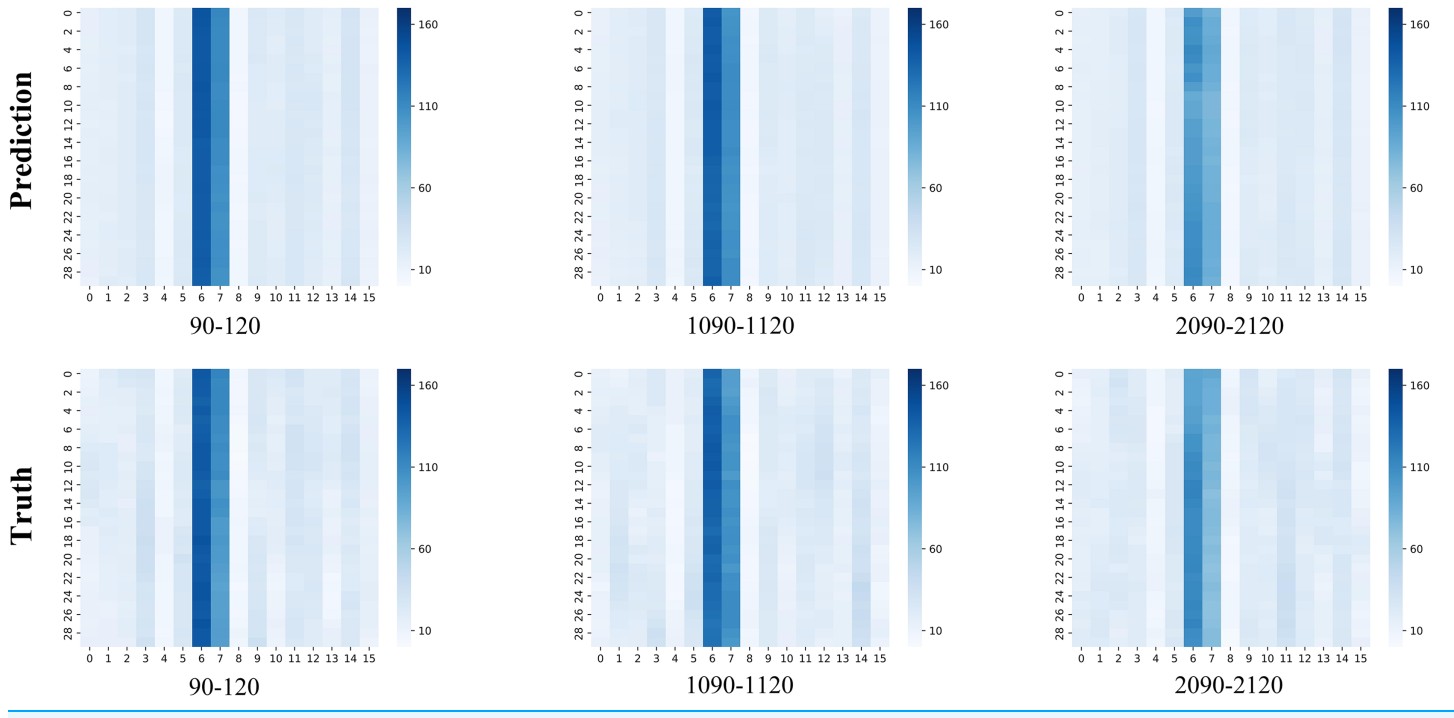

**Figure 3 Comparison of true OD *vs.* predicted OD.**

performance on MAE, MSE, and RMSE when the learning rate is set to 1e−5, and second-best performance for SMAPE.

L2 regularization is a common technique employed to prevent overfitting by incorporating a penalty term proportional to the squared L2 norm of the weight matrix

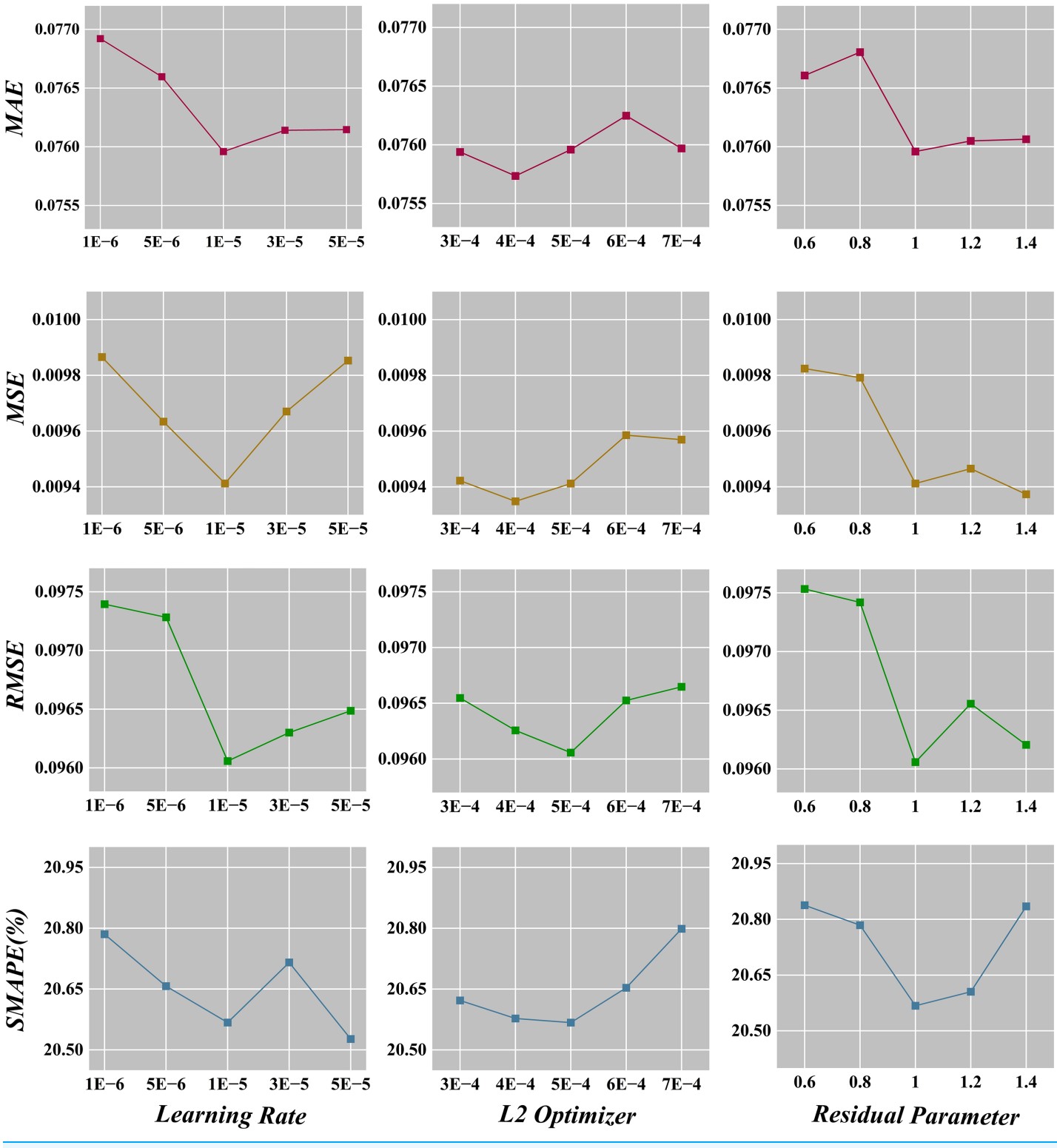

**Figure 4 Sensitive analysis.**

into the loss function, thereby limiting model complexity. Figure 4 shows that the fluctuations of all four metrics are relatively minor as the L2 regularization coefficient in the optimizer changes, suggesting that the model is not highly sensitive to adjustments in this specific parameter. Therefore, a default value of 5e−4 can be appropriately chosen based on its generally stronger model performance across the investigated range.

The residual ratio parameter represents the extent to which the original input signal is combined with the signal processed by GCN layers. As depicted in the relevant chart, no clear pattern emerges in the four metrics as the residual ratio shifts. Nevertheless, it can be observed that the model performs favorably with other settings when the residual ratio is fixed at 1, thus supporting the selection of a default value of 1, which aligns with the commonly used residual link ratio in residual networks.

## CONCLUSIONS

In this study, we utilize the GraphResLSTM model to estimate OD pairs on road average speed data. By integrating an entropy-based weight allocation strategy and using the weighted TOPSIS method to select representative road, our deep learning architecture efficiently captures spatial-temporal dependencies. GraphResLSTM model fully leverages the topological structure information of the road network, enabling it to learn long-term dependency patterns from historical traffic conditions. Additionally, it ensures stable convergence even in deeper architectures, enhancing its ability to capture intricate traffic dynamics. The experimental results have demonstrated that the GraphResLSTM model excels in OD prediction tasks, offering a state-of-the-art and efficient solution with exceptional accuracy for traffic management.

Through the use of road average speed data rather than traditional traffic flow data for forecasting, the GraphResLSTM model is able to more accurately capture the spatiotemporal characteristics of urban traffic systems, leading to more precise OD predictions. For urban traffic management departments, this translates into advanced planning and adjustment of traffic signals, optimization of public transportation schedules, and rational allocation of emergency resources, thereby effectively mitigating traffic congestion during peak hours and improving the overall operational efficiency of the city's traffic system. Additionally, this method decreases the need for costly sensor infrastructure, reduces data collection expenses, and streamlines the data preprocessing required for model training, thus saving time and human resources and making intelligent transportation systems more economically viable.

Constrained by time and equipment, the study primarily relies on simulated data, may not fully capture the complex traffic patterns and unforeseen events found in real-world scenarios, such as traffic accidents, extreme weather conditions, or unexpected traffic alterations due to large-scale events. Future research could incorporate real-world traffic data, including historical traffic flow data, weather forecast information, and notifications about special events, to enhance the model's responsiveness to real-world occurrences. This comprehensive approach aims to provide a deeper understanding of the various factors influencing OD predictions. Additionally, traffic data from multiple cities will be

utilized for model training and validation, assessing the model's applicability and robustness across diverse geographic settings and socio-economic contexts.

### Funding

This study is supported by grants from the National Social Science Foundation of China (grant number 20BGL001), which is required to be unique. The funders had no role in study design, data collection and analysis, decision to publish, or preparation of the manuscript.

### Grant Disclosures

The following grant information was disclosed by the authors:
National Social Science Foundation of China: 20BGL001.

### Competing Interests

The authors declare that they have no competing interests.

### Author Contributions

- Guangtong Hu conceived and designed the experiments, performed the experiments, analyzed the data, performed the computation work, prepared figures and/or tables, and approved the final draft.
- Jun Zhang conceived and designed the experiments, authored or reviewed drafts of the article, and approved the final draft.

### Data Availability

The raw measurements are available in the Supplemental Files.

### Supplemental Information

Supplemental information for this article can be found online at http://dx.doi.org/10.7717/peerj-cs.2709#supplemental-information.

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
