# Peer review of "Origin-destination prediction from road average speed data using GraphResLSTM model"

_PeerJ Computer Science, doi:10.7717/peerj-cs.2709_

## Round 0.1 · original submission · Major Revisions

This was a resubmission of a prior 'open rejection' decision.

Please study the comments by the 2nd reviewers, respond to them as appropriate, and revise your paper accordingly. I will ask the reviewer to consider your response and then decide on the outcome. Thanks for your continued interest in the journal.

·

Basic reporting

The authors have addressed all my concerns, and the manuscript is ready for publication.

Experimental design

None.

Validity of the findings

None.

Additional comments

None.

Reviewer 2 ·

Basic reporting

This paper introduces a hybrid model named GraphResLSTM that integrates Graph Convolutional Networks (GCN), Residual Networks (ResNet), and Long Short-Term Memory (LSTM) networks for predicting Origin-Destination (OD) matrices using road average speed data. The authors employed SUMO simulation software to generate traffic data and applied the Entropy Weight Method combined with the Technique for Order Preference by Similarity to Ideal Solution (TOPSIS) for selecting key road segments. Experimental results demonstrate that the GraphResLSTM model outperforms baseline models in several metrics, including MAE, MSE, and RMSE. The topic of the manuscript is interesting. However, there are several areas that require further clarification and expansion for the manuscript to be suitable for publication.
Detailed comments are shown below:
1. Provide a more detailed explanation of how each component of the GraphResLSTM (GCN, ResNet, LSTM) contributes to the overall performance. For instance, explain why GCN is well-suited for spatial dependencies and how ResNet prevents gradient issues.
2. Incorporate experiments with real-world traffic data to validate the model's practical applicability. Simulation data alone limits the reliability of the findings.
3. Expand on the preprocessing steps, including a clear description of the Entropy Weight Method and TOPSIS processes. Justify the selection of the six key metrics used for road segment evaluation.
4. The authors should strengthen the literature review, and the latest references should be supplemented. The following references are suggested for consideration: [1] Audio related quality of experience evaluation in urban transportation environments with brain inspired graph learning;[2] GIS aided sustainable urban road management with a unifying queueing and neural network model;[3] AI-empowered speed extraction via port-like videos for vehicular trajectory analysis;[4]Resilience Analysis of Urban Road Networks Based on Adaptive Signal Controls: Day‐to‐Day Traffic Dynamics with Deep Reinforcement Learning
5. Discuss the results with more depth. For instance, elaborate on why GraphResLSTM achieves lower MAE and MSE compared to other models and what implications these metrics have for real-world OD prediction.
6. Strengthen the conclusion by emphasizing the practical implications of the findings. In the future work section, suggest specific directions, such as integrating additional data sources (e.g., weather or incident reports) or testing in diverse cities with varying traffic patterns.
7. It is advisable to review the article for grammar and spelling errors, ensuring clear and accurate language. Additionally, avoid using vague or ambiguous terms.

Experimental design

N/A

Validity of the findings

N/A

---

## Round 0.2 · accepted · Accept

Thanks for undertaking the revision and doing a good job. I am happy to recommend the paper for publication.

Reviewer 2 ·

Basic reporting

The authors have addressed most of my concerns, but further minor revisions are still required. Literature review should be further enhanced.

Experimental design

The authors have addressed most of my concerns, but further minor revisions are still required. Literature review should be further enhanced.

Validity of the findings

The authors have addressed most of my concerns, but further minor revisions are still required. Literature review should be further enhanced.

Additional comments

The authors have addressed most of my concerns, but further minor revisions are still required. Literature review should be further enhanced.